# Adversarial Testing in LLMs: Insights into Decision-Making Vulnerabilities

## Abstract

As AI systems, particularly Large Language Models (LLMs), rapidly advance towards surpassing human cognitive capabilities, ensuring their alignment with human values and safety standards emerges as a formidable challenge. This study addresses a crucial aspect of superalignment by investigating the decision-making capabilities and adversarial vulnerabilities of LLMs, focusing on GPT-3.5, GPT-4 and Gemini-1.5, within structured experimental settings that mimic complex human interactions. We applied an adversarial framework to two decision-making tasks—the two-armed bandit task and the Multi-Round Trust Task (MRTT)—to test the vulnerabilities of LLMs under adversarial conditions. In the bandit task, the adversary aimed to induce the LLM's preference for the predefined target action with the constraint that each action must be assigned an equal number of rewards. For the MRTT, we trained two types of adversaries: one aimed at maximizing its own earnings (MAX) and the other focused on maximizing fairness (FAIR). GPT-4 and Gemini-1.5 showed a bias toward exploitation in the bandit task, prioritizing early-established strategies, which made them predictable and vulnerable to manipulation. GPT-3.5, while more exploratory in the bandit task, demonstrated more risk-seeking behavior in the MRTT, leading to increased vulnerability in interacting with the MAX adversary. Notably, Gemini-1.5 excelled in the MRTT, adapting effectively to adversaries and outperforming both GPT-3.5 and GPT-4 by balancing risk and cooperation with its adversaries. By presenting a specific set of tasks that characterizes decision-making vulnerabilities in LLM-based agents, we provide a concrete methodology for evaluating their readiness for real-world deployment. The adversarial framework proved a powerful tool for stress-testing LLMs, revealing the importance of ensuring that AI models are both robust against adversarial manipulation and responsive to fairness cues in complex, dynamic environments.

## 1 Introduction

In recent years, the landscape of Artificial Intelligence (AI) has been reshaped by the emergence of Large Language Models (LLMs) such as Generative Pre-trained Transformers (GPT). These models have demonstrated remarkable performance across various applications, from natural language processing to complex problem-solving tasks. As LLMs are increasingly integrated into decision-making processes in sectors such as healthcare (Karabacak & Margetis, 2023), finance (Krause, 2023), and autonomous systems (Sha et al., 2023), they augment human capabilities in data analysis, forecasting, and strategic planning. However, as LLMs advance toward cognitive capabilities that may surpass human functions, ensuring that their operations are aligned with human values and safety standards becomes crucial. This challenge, often referred to as "superalignment", is particularly critical in domains with significant ethical and practical implications. Achieving superalignment necessitates a deep understanding of the decision-making processes of LLMs, including the factors influencing these decisions and potential biases they barbor (Rahwan et al., 2019). Insights into these processes are essential for mitigating risks and ensuring that AI systems operate in a way that is predictable, reliable, and aligned with ethical standards.

While significant efforts have focused on improving LLM architectures and optimizing hyperparameters, there is growing recognition that evaluating these models' decision-making capabilities requires a more interdisciplinary approach that goes beyond traditional performance metrics. By employing

methodologies from cognitive psychology and game theory, researchers can treat LLMs as active participants in structured psychological experiments, providing a more comprehensive assessment of their cognitive abilities compared to human norms (Hagendorff, 2023). For instance, the use of psychology-inspired tests has revealed cognitive biases and different problem-solving approaches of LLMs that extend beyond traditional performance-based metrics, highlighting their limitations in deeper reasoning and causal understanding. A pioneering study by Binz et al. (Binz & Schulz, 2023) assessed GPT-3's cognitive abilities through a series of cognitive experiments, revealing that while GPT-3 can generate superficially appropriate responses, its decision-making falters with deeper reasoning or causal understanding. Subsequent studies have evaluated LLMs' cognitive performance from different aspects, such as analogical reasoning, theory of mind, and problem-solving (Webb et al., 2023; Kosinski, 2023; Orrù et al., 2023). Hagendorff et al. (Hagendorff et al., 2022; 2023) explored intuitive and deliberative thinking (System 1 and System 2 processes) in assessing LLMs' behaviour and reasoning biases. This series of studies identified a significant evolution in LLM capabilities from pattern recognition to human-like reasoning and decision-making. The exploration has been extended into social exchange scenarios where strategic thought and game-theoretic reasoning are required. It was found that while LLMs can learn and apply strategies, they struggle with complex strategies like forgiveness and deception, and generalizing across different contexts (Fan et al., 2024; Akata et al., 2023; Huang et al., 2024).

This line of research has shown that, despite LLMs' advancements, they still struggle with human-like strategic reasoning, particularly in complex social and decision-making scenarios . The application of psychological approaches in these studies highlights the potential of interdisciplinary research in advancing AI. Leveraging principles of human cognition could enhance LLM behavior and uncover susceptibilities to biases and manipulations (Yao et al., 2024). Building on these insights, this paper proposes to use a novel adversarial framework (Dezfouli et al., 2020) specifically designed to assess decision-making vulnerabilities in LLMs. Rather than focusing solely on performance in specific tasks, this framework enables systematic probing of LLMs' decision-making processes under adversarial interactions, providing a structured way to assess how LLMs adapt—or fail to adapt—when faced with dynamic, strategic opponents.

We demonstrate the utility of the adversarial framework through experiments on GPT-3.5, GPT-4, and Gemini-1.5 across two decision-making tasks: the two-armed bandit task and the Multi-Round Trust Task (MRTT). These experiments validate the framework's ability to uncover model-specific vulnerabilities and provide insights into LLM decision-making mechanisms. Designed to be adaptable, this framework is applicable to a wider range of decision-making scenarios and LLM architectures, offering a versatile tool for future AI safety research. Notably, the aim of this paper is not to comprehensively assess all LLMs but to present a novel approach for evaluating decision-making alignment in intelligent agents. The primary contributions of this paper are as follows:

- **Adversarial Framework for LLM Evaluation.** We introduce a structured adversarial framework to reveal vulnerabilities in LLM decision-making, highlighting biases such as exploitation and risk-seeking behaviors under adversarial influence.

- **Superalignment and Ethical Decision-Making.** This study addresses the challenge of aligning advanced LLMs with human values and safety standards by applying cognitive and game-theoretic insights into LLM decision-making processes.

- **Model-specific vulnerabilities.** Our experiments reveal specific vulnerabilities in the models, such as exploitation bias in GPT-4 and Gemini-1.5 during the bandit task and risk-seeking behavior in GPT-3.5. Gemini-1.5 outperformed the other models in the MRTT, showing better adaptability and balance between risk and cooperation, which suggests that certain LLMs can be trained to maintain fairness and adaptability in dynamic interactions.

## 2 METHOD

### 2.1 THE ADVERSARIAL FRAMEWORK

The adversarial framework is structured in a multi-phase process (see Fig 1). In the initial phase, we collect behavioral data from GPT-3.5, GPT-4 and Gemini-1.5 during a decision-making task (Fig. 1*A*) In each interaction $n$, on trial $t$, the LLM receives a learner reward ($r_t^n$) based on its previous

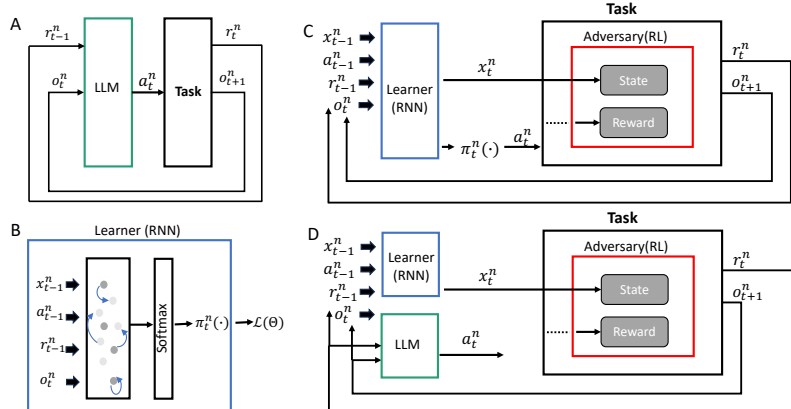

Figure 1: The adversarial framework (adapted from Dezfouli et al. (2020)). **(A)** The interaction of the LLM with the task. Each simulation cycle begins with the LLM receiving a learner reward ($r_{t-1}^n$) for the prior action along with a new observation ($o_t^n$) from the environment. Based on this, the GPT executes an action ($a_t^n$), and the cycle repeats with the environment providing updated rewards and observations. **(B)** The LLM's actions are modelled by a RNN with parameters $\Theta$. Inputs to the RNN include the previous action ($a_{t-1}^n$), the most recent learner rewards ($r_{t-1}^n$), and the current observations ($o_t^n$), along with the RNN's last internal state ($x_{t-1}^n$). After receiving the inputs, the RNN updates its internal state and predicts the next action using a softmax layer ($\pi_t^n$). These predictions are then compared with the actual actions taken by the LLM and evaluated with a loss function ($\mathscr{L}(\Theta)$) in order to train the model. The trained model is called the learner model. **(C)** The adversary is an RL agent, which is trained to manipulate the decision-making environment of the learner model. Utilizing the latest internal state ($x_t^n$) of the learner model, which encapsulates its cumulative learning experiences, the adversary determines the learner reward ($r_t^n$) and the next observation ($o_{t+1}^n$) to be delivered to the learner model. This strategic input is designed to steer the learner model's subsequent actions ($a_t^n$) toward achieving the adversary's predefined objectives. The adversarial reward (*Reward*), which is used to train the adversary, depends on the alignment between the action taken by the learner model ($a_t^n$) and the adversary's objectives. **(D)** Using the trained adversary and the learner model for generating adversarial interactions with the LLM. In each simulation $n$, the LLM processes the rewards ($r_{t-1}^n$) and observations ($o_t^n$) from the adversary, responding with actions ($a_t^n$) that update the learner model's internal state ($x_t^n$). This state is then sent to the adversary to determine the learner's reward for the action ($a_t^n$) and the next observation ($o_{t+1}^n$). This cycle continues till the end of the task.

action ($a_{t-1}^n$) and the current observation ($o_t^n$), which is the feedback text. The LLM then takes the next action, $a_t^n$. The process repeats with the LLM receiving the learner reward of the action chosen ($r_t^n$) and the next observation ($o_t^n$).

The data collected is then used to train a learner model (determined by parameters $\Theta$) to predict the LLM's next action in the decision-making task (Fig. 1*B*). The learner model consists of an RNN and a softmax layer, which has shown sufficient capacity to capture the patterns and tendencies in the decision-making entity's choices (Dezfouli et al., 2019b;a). The inputs to the RNN include the previous action ($a_{t-1}^n$), the learner reward $r_{t-1}^n$,, and the current observations from the task ($o_t^n$) along with the previous internal state of the RNN ($x_{t-1}^n$). The internal state ($x_{t-1}^n$) is recurrently updated in each trial based on the inputs and is then mapped to a softmax layer to predict the next action $\pi_t^n(\cdot)$. These predictions are then compared with the actual actions taken by the LLM using a loss function ($\mathscr{L}(\Theta)$), which is used to train the model.

The next phase involves developing an RL agent as the adversarial model (Fig. 1*C*). This model is trained to interact with the learner model to identify and exploit weaknesses in the decision-making patterns. By manipulating inputs or altering the decision-making environment, the RL adversary aims to influence the outcomes in a way that demonstrates the vulnerabilities of the decision-making process. It uses the internal state of the learner model ($x_t^n$ for simulated learner $n$) as the state of the environment to decide the learner reward $r_t^n$ and next observation $o_t^n$ to be provided to the learner.

The learner model takes its next action and this cycle continues with the new state of the learner model ($x_{t+1}^n$) being passed to the adversary. The adversary's policy is trained to maximize cumulative adversarial rewards using Deep Q-learning (Mnih et al., 2015).

In the final phase, the trained adversary and learner model interact with the LLM. The learner model does not choose actions, but receives the actions made by the LLM ($a_t^n$) as input and tracks their learning history using its internal state $x_t^n$. In turn, $x_t^n$ and the actual action taken by the LLM are fed to the adversary to decide the learner reward $r_t^n$ and next observation $o_{t+1}^n$, which the LLM will use to choose their next action $a_{t+1}^n$. The same input, along with the LLM's action, is sent to the learner model. This cycle continues until the end of the task.

## 2.2 THE TWO-ARMED BANDIT TASK

We applied the framework to develop adversaries for GPT-3.5 and GPT-4 on two decision-making tasks: the two-armed bandit task and the (MRTT). The bandit task is a repeated, two-alternative forced-choice task based on the bandit task introduced by Dan & Loewenstein (2019). The task includes 100 trials, where the LLM selects between two options and receives instant feedback indicating a reward or no reward after each decision. The adversary assigns rewards to both potential actions with the constraint that each action receives an equal number of potential rewards (25 times). This experiment aims to subtly influence GPT's preferences and evaluate the adversary's effectiveness under these constraints.

## 2.3 THE MULTI-ROUND TURST TASK

The MRTT is designed as a structured interaction between two participants: the "investor" and the "trustee" (Brooks King-Casas et al., 2005; McCabe et al., 2003). The task contains 10 sequential rounds, with the investor initially receiving 20 monetary units at the outset of each round. The investor decides how much of this endowment to allocate to the trustee. The experimenter triples the invested amount and sends it to the trustee, who can then return any portion of the received amount as repayment. Cumulative earnings for each participant are calculated by summing their respective gains from all rounds.

## 3 RESULTS

### 3.1 THE TWO-ARMED BANDIT TASK

The objective of this experiment was to examine how the LLMs respond to rewards based on their choices in the two-armed bandit task and to assess if an adversary can manipulate their preferences toward a predetermined target action. Data were generated from GPT-3.5, GPT-4 and Gemini-1.5 by providing prompts (as illustrated in Fig 2A) corresponding APIs. The system message established the context for the LLM's behaviour and decision-making process within the simulation. In this scenario, the LLM plays the role of a space explorer deciding between visiting two planets, X or Y, to find gold coins. Each trial's prompt asks the model which planet to visit, with responses and outcomes from previous trials included. GPT-3.5 and Gemini-1.5 were simulated 200 times, and GPT-4 was simulated 100 times, with each simulation consisting of 100 trials. The reward probability for the two options were the same, both of which were 25%, and the target option was defined as Planet X. We used the dataset provided by Dan & Loewenstein (2019) as a benchmark for human performance on the two-armed bandit task.

**Behavioural Analysis** Fig 2B illustrates the decision-making behaviours of humans and LLMs across trials in the two-armed bandit task. Blue and red circles indicate the target or the non-target Planet was selected and the corresponding vertical lines represent the selected option yielded rewards. The human data shows a dynamic pattern of switching between the target and non-target options, with behaviour adapting over the course of the trials. When participants encounter consecutive trials without rewards, they tend to reassess their choice, often switching to the other option in subsequent trials. In contrast, the LLMs exhibit more rigid and predictable behavior patterns. Particularly, GPT-4 and Gemini-1.5 display an initial phase of exploration, but quickly converge on one option once a reward is obtained. GPT-3.5 also tends to commit more strongly to one option, but it displays more

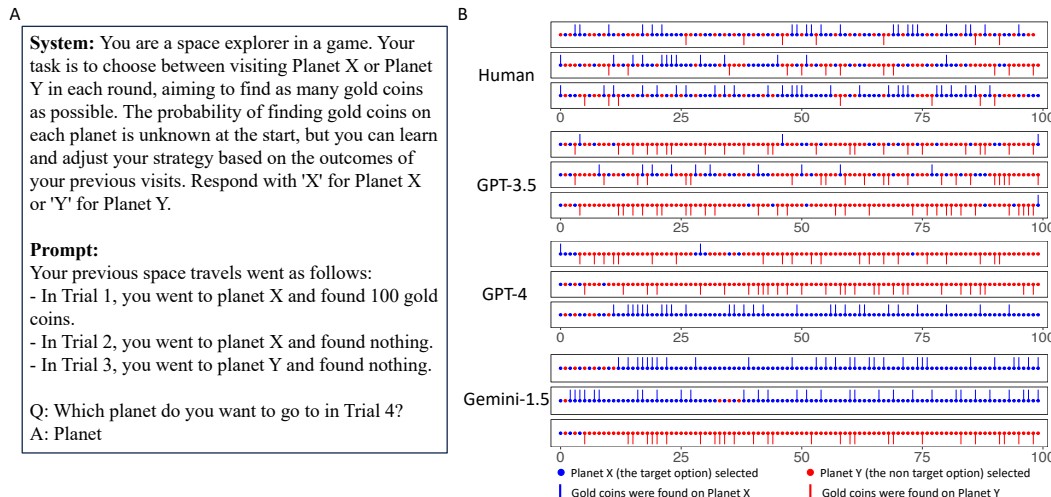

Figure 2: **A:** Example prompt for one trial in the bandit task for the LLMs. **B:** Behavioural pattern in trials of three random human participants and three sample simulations for each of the LLMs.

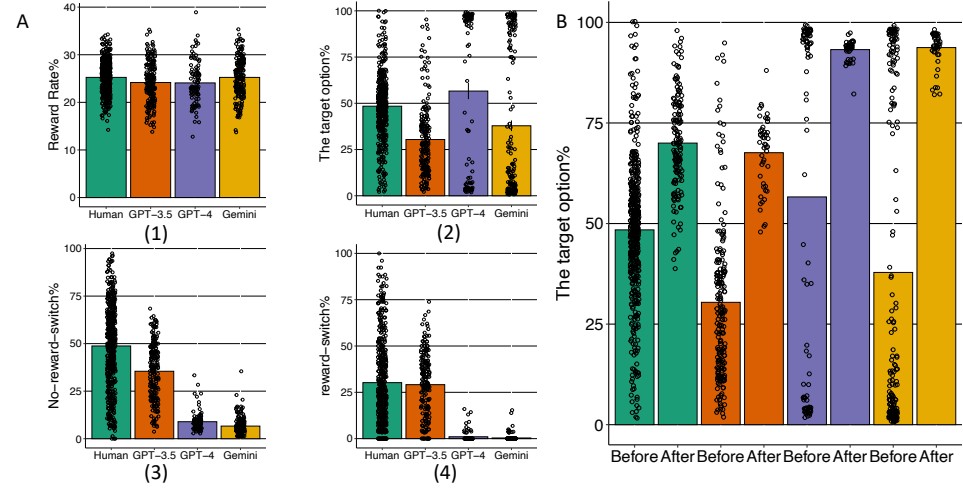

Figure 3: **A:** LLMs' behaviour compared to human behaviour on average of each simulation (or individual), measured by reward rate, percentage choosing the target option, no-reward-switch rate, and reward-switch rate. **B:** The performance of human and the LLMs, which is measured by the percentage of the target action selection before and after adversarial influence.

flexibility than GPT-4 and Gemini-1.5 in later trials, as evidenced by occasional switches to the other option, particularly after experiencing multiple trials without rewards.

Fig 4 compares LLMs and human performance on the bandit task across four metrics: : (1) reward rate, (2) percentage choosing the target option, (3) no-reward-switch rate, and (4) reward-switch rate. Humans earned significantly higher rewards than GPT-3.5 (mean difference: 1.074, $p = 0.005$) and GPT-4 (mean difference: 1.163, $p = 0.030$), while Gemini-1.5 achieved similar rewards to humans (mean difference: $0.004$ $p = 0.962$). GPT-3.5 and Gemini-1.5 consistently preferred Planet Y over Planet X (GPT-3.5: $t(201) = -14.14, p < 0.001$, Gemini-1.5: $t(201) = -13.94, p < 0.001$), while GPT-4 displayed consistent preferences for either, depending on the simulation. Humans, however, showed more diversified choices ($t(483) = -1.76, p = 0.07$), indicating varied exploration strategies. In terms of no-reward-switch rate (changing choices after negative feedback), all LLMs switched less frequently than humans, with GPT-4 and Gemini-1.5 switching even less than GPT-3.5

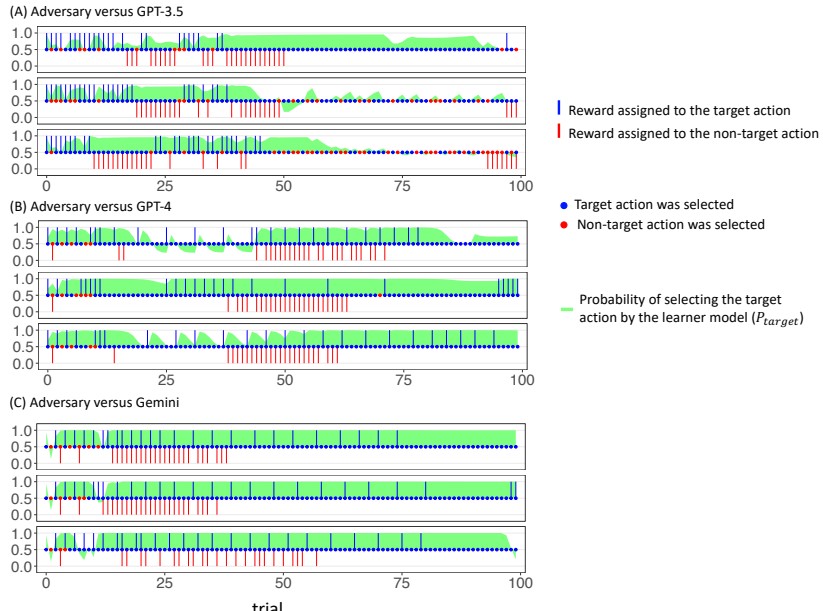

Figure 4: Three sample simulations of the trained adversaries against GPT-3.5, GPT-4, and Gemini-1.5, respectively. The plot presents the strategies used by the adversaries and the responses of the LLMs. **(A)** Adversary versus GPT-3.5. **(B)** Adversary versus GPT-4. **(C)** Adversary versus Gemini-1.5

(GPT-4 vs GPT-3.5: $p < 0.001$; Gemini-1.5 vs GPT-3.5: $p < 0.001$ ), implying that GPT-4 and Gemini-1.5 change their decision-making behaviour less in response to losses compared to GPT-3.5. For reward-switch rate (changing choices after rewards), humans and GPT-3.5 were more adaptable than GPT-4 and Gemini-1.5, which exhibited the lowest frequency and variability in reward-switch rates (GPT-4 vs. GPT-3.5: $p < 0.001$; GPT-4 vs. human: $p < 0.001$; Gemini-1.5 vs. GPT-3.5: $p < 0.001$; Gemini-1.5 vs. human: $p < 0.001$). These suggests that GPT-4 and Gemini-1.5 are more rigid in their decision-making, sticking to initial preferences and adapting less to losses or rewards compared to humans and GPT-3.5.

**Adversarial Analysis** Simulated data from GPT-3.5, GPT-4, and Gemini-1.5 were used to train a learner model for each LLM. We then trained deep Q-learning models as adversaries to exploit these learners (see Appendix for details of the training parameters). The adversary's goal was to induce the learner to select a predetermined target action, with a constraint limiting rewards to 25 per action. The trained adversaries were then evaluated against the LLMs (Fig. 1 *D*), based on the proportion of trials in which the chosen action aligned with the target action (Planet X) before and after adversarial influence. As shown in Fig. 3*B*, GPT-3.5 and Gemini-1.5 initially favored the non-target action (Planet Y), but adversarial influence increased their target selection from 30% to 68% and from 38% to 94%, respectively. GPT-4 showed a mixed preference for either option, averaging a 56% target selection rate, which was increased to 93% under adversarial manipulation. These increases indicate effective adversarial control of LLM decision-making.

Finally, we sought to understand the strategy used by the adversaries and the responses of the LLMs' when subjected to adversarial strategies. Fig. 4 presents sample adversarial strategies and LLM responses. The green shaded area represents the probability of the learner model choosing the target option. A higher green area across trials implies the adversary consistently influences the learner model (and hence the LLMs) towards the target action. GPT-3.5's adversary started by rewarding the target action and then "burned" non-target rewards when it predicted the non-target option would not be chosen. When GPT-3.5 chose the non-target action, the adversary realigned preferences by rewarding the target action. Despite the initial stability, GPT-3.5 began switching actions when target rewards were depleted. This exploratory action suggested a robust decision-making characteristic of GPT-3.5 that integrates new information continuously, assessing the potential

benefits of diverging from established preferences. For GPT-4 and Gemini-1.5, their adversaries assigned disproportionately rewards to the target action at the start to establish baseline preference, then intermittently to maintain it, especially when the model's selection of the target action began to wane. Unlike GPT-3.5, GPT-4 and Gemini-1.5 consistently preferred the target action once established. Their adversaries could easily "burn" non-target rewards without being noticed by these two LLMs.

## 3.2 MULTI-ROUND TRUST TASK

In the MRTT, the LLM plays the investor and the adversary plays trustee. The adversary's decisions, representing the proportion of money sent back to the investor, are categorized into five actions (0, 25, 50, 75, and 100%). The objective of the adversary was to influence the LLMs' investment decisions to align with its goals. We developed and trained two types of adversaries for each LLM: MAX and FAIR. The MAX adversary aimed to maximize its total gain over 10 rounds, adopting a competitive strategy. The FAIR adversary sought to distribute earnings evenly between itself and the LLM, adopting an equitable strategy. This dichotomy in objectives allowed us to examine how the LLMs respond to different adversarial strategies, revealing their capabilities in complex social exchanges and decision-making processes.

**Behavioural Analysis** We firstly collected data from GPT-3.5, GPT-4, and Gemini-1.5 playing against with a random trustee (also called as random adversary in the following, i.e. the trustee selects repayment action uniformly at random). The prompts for interacting with the LLM are shown in Fig. 5 *A*. The system message sets the scenario for the LLM and decision-making process. In each round, the LLM received a summary of previous rounds, including the amount the LLM invested, the consequential action the trustee took, and the total earnings from the transaction in each round. Following this summary, the LLM was asked about its investment decision for the current round. The three LLMs were all simulated for 200 times, with 10 rounds of interaction with a random adversary.

We assessed the comparative performance dynamics between human subjects (from Dezfouli et al.'s study) and LLMs on the MRTT. Fig 5*B* illustrates how varying repayment amounts influence investment decisions in subsequent rounds for humans and the three LLMs. All subjects demonstrated a trend where the investment in the current trial increases as the repayment in the previous trial increases, indicating reinforcement learning behaviour. However, GPT-4 consistently made the most conservative investments (not more than 10 units) across all repayment intervals, especially in the higher repayment brackets, where its investments were significantly lower than those of humans and the other two LLMs. Gemini-1.5 exhibited the most pronounced sensitivity to repayment feedback among all subjects, significantly increasing its investment when the repayment in the previous trial was high.

**Adversarial Analysis** Using data from the random condition, we trained the learner model, which was then used to train two types of adversaries: MAX (aiming to maximize earnings over 10 rounds) and FAIR (aiming to balance earnings between the trustee and the investor). We simulated 50 times for each LLM interacting with their adversaries. 5*C* compares the overall earnings of the subjects and their adversaries. Fig. 6 depicts dynamic interactions over trials, in which the left panel illustrates how the adversaries adjust their repayment behavior, and the right panel shows how the investors adjust their investment based on the repayment feedback over trials. Human data is from Dezfouli et al., including 155 subjects with the FAIR adversary and 209 with the MAX adversary.

All MAX adversaries maintained relative higher investment levels compared to the other two adversaries from their counterparts (humans or LLMs), indicating effective exploitation of the decision-making patterns of their counterparts. Humans and GPT-3.5, in particular, exhibited a tendency to maintain high investments despite receiving low repayments, suggesting optimism, higher risk tolerance, or susceptibility to MAX tactics. This allowed their adversaries to extract the highest earnings (273 and 377 units), creating the largest earning gap between the two. In contrast, GPT-4 adopted a far more conservative strategy, minimizing its investments even when the adversary offered high repayments. Its MAX adversary had to offer intermittent high repayments to keep GPT-4 engaged, preventing substantial earnings from GPT-4 by the end of the task (the difference between trustee earning and investor earning is negative as shown in Fog 5) *C*. Gemini-1.5, though more willing to invest than GPT-4, displayed a more balanced approach to risk and investment. While Gemini-1.5 maintained higher investments throughout the trials, its adversary struggled to extract

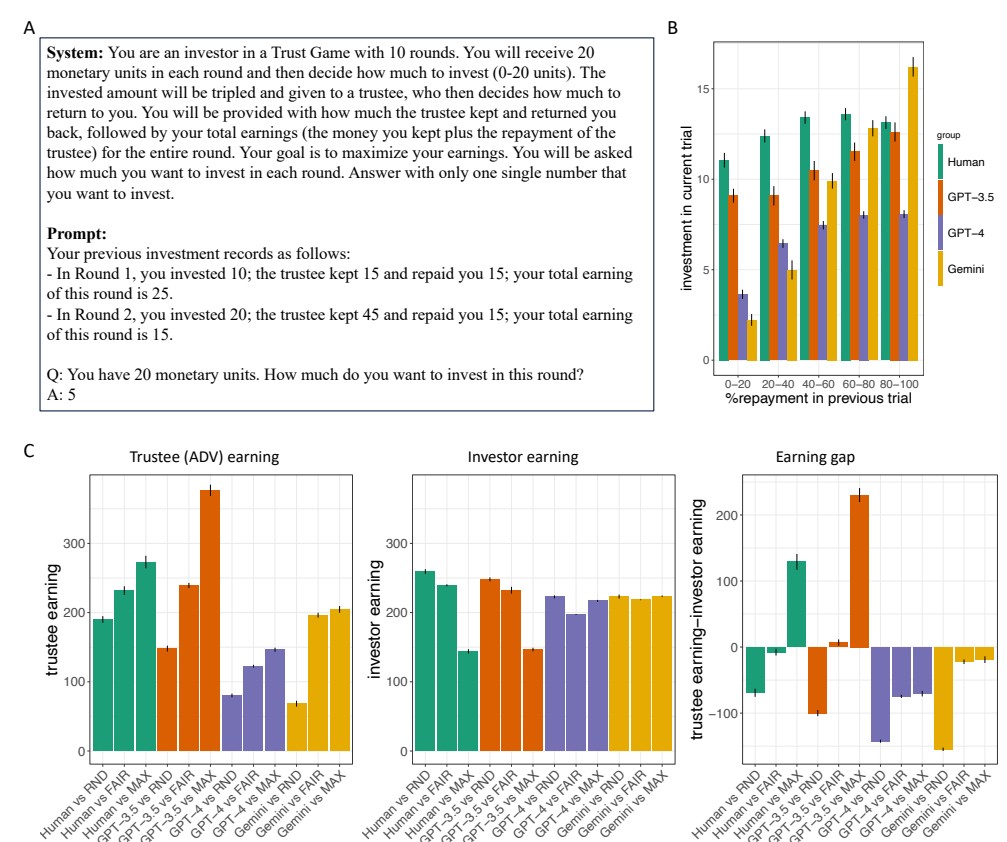

Figure 5: **A:** Example prompt for one round in the MRTT as submitted to GPT-3.5 or GPT-4. **B:** GPT-3.5 and GPT-4's behaviour versus human behaviour playing against a random trustee in the MRTT. The investment in a round is a function of the proportion of the investment repaid by the trustee in the previous round. **C** The total amount earned by the trustee (the adversary) and the investor (human, GPT-3.5 or GPT-4) and the absolute gap between them in different conditions.

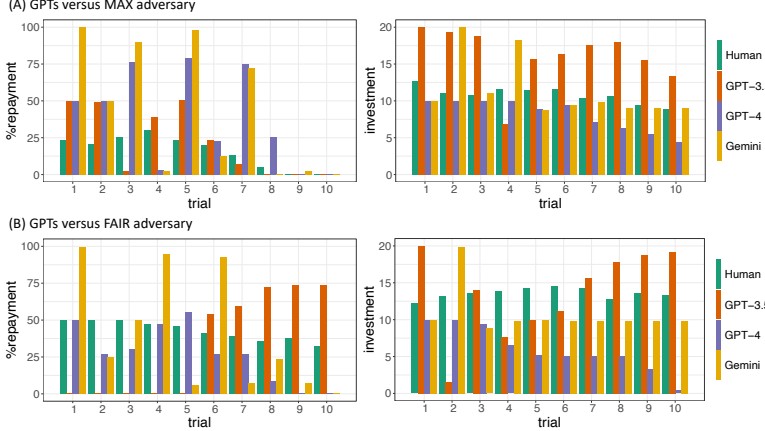

Figure 6: The percentages of investment and repayment in each trial for MAX and FAIR adversaries.

higher earnings due to the need to offer consistently high repayments to keep Gemini-1.5 engaged and did not manage to obtain more earnings than Gemini-1.5 either (MAX adversary: 205 units vs Gemini-1.5: 224 units).

In the interaction with the FAIR adversary, human participants and their FAIR adversary achieved a win-win with high earnings, i.e. both human and its adversary achieved high earnings and the gap between them was minimised. Humans recognized fairness cues and maintained high investment despite the adversary's repayment rate decreasing since the third trial. GPT-3.5 displayed a lack of sensitivity to repayments and maintained relatively high investments despite receiving little or no repayment in the early trials, suggesting a risk-seeking tendency. Its adversary had to repay higher amounts in the later trials to ensure fair outcomes for the two sides. In contrast, GPT-4 adopted a highly conservative, risk-averse strategy, steadily reducing its investments over time, even when moderate repayments were offered, resulting in lower earnings for both GPT-4 and its adversary. Meanwhile, Gemini-1.5 responded to early high repayments with increased investment and stabilized at a consistent level in the following trials, demonstrating greater adaptability and resilience to its adversary's attempts at fostering a fair environment.

## 4  DISCUSSION

Our experiments reveal important insights into the decision-making processes of the LLMs compared to humans, expecially in their responses to adversarial strategies in dynamic environments. These findings highlight both the vulnerabilities and strengths of LLM decision-making, with significant implications for real-world applications. The adversarial framework employed in this study was central to understanding the LLMs' responses to changing conditions and adversarial tactics. By placing LLMs in scenarios where they interacted with adversaries in the bandit task and the MRTT, we were able to observe how different models reacted in situations that involves choice engineering and social exchanges. The adversarial setup revealed nuanced patterns of behavior that would not be apparent through traditional testing.

In the bandit task, most human participants exhibited a balanced approach between exploration and exploitation, dynamically adjusting their strategies to capitalize on rewards more effectively (Wilson et al., 2014). In contrast, GPT-4 and Gemini-1.5 showed a stronger tendency to exploit a single option, likely driven by algorithmic optimization of past rewards Binz & Schulz (2023); Nguyen & Satoh (2024). The adversarial framework revealed how this computational bias toward exploitation made their decision-making highly predictable, exposing their vulnerability to manipulation when interacting with adversarial strategies. While such exploitation minimizes risk, it limits these models' ability to adapt to changing conditions, leading to potential inefficiencies in dynamic environments. GPT-3.5, on the other hand, demonstrated greater flexibility, as the adversarial framework exposed its tendency to test alternative strategies, especially in response to non-rewarding outcomes. However, this exploratory behavior also made GPT-3.5 more vulnerable to exploitation in the MRTT, where its risk-seeking approach was exploited by the MAX adversary, leading to the largest earnings gap. The framework helped clarify this trade-off: exploration opens opportunities in uncertain environments but can also increase the risk of exploitation as it may lead to testing riskier strategies, while exploitation biases provide stability but reduce adaptability.

In real-world applications—such as finance, healthcare, and autonomous systems—AI systems must strike a careful balance between exploration and exploitation to thrive in dynamic, unpredictable environments. AI models that favor exploitation, as seen with GPT-4 and Gemini-1.5, are prone to predictable behavior, limiting their ability to respond effectively to adversarial tactics or new opportunities. Conversely, while GPT-3.5's exploratory tendencies allowed it to engage more flexibly with its environment, the framework revealed its susceptibility to adversarial exploitation. These insights emphasize the value of adversarial testing in stress-testing AI decision-making.

The MRTT further demonstrated the power of the adversarial framework in uncovering differences in LLM behavior when navigating complex economic exchanges (Xie et al., 2024). The FAIR adversary aimed to foster cooperation, but the conservative strategy adopted by GPT-4 limited its ability to engage fully, despite the adversary's efforts to encourage greater investment (Rafailov et al., 2024). In contrast, Gemini-1.5's balanced approach allowed it to adapt dynamically to both the MAX and FAIR adversaries, capitalizing on reciprocal fairness while avoiding excessive exploitation. This adaptability enabled Gemini-1.5 to outperform both GPT-4 and GPT-3.5 in maximizing gains while

remaining resilient against exploitation. In adversarial settings like cybersecurity or competitive business environments, the ability of AI systems to adjust dynamically is critical for success. For instance, models like GPT-4, which prioritize stability over flexibility (Akata et al., 2023; Huang et al., 2024), risk missing opportunities for reciprocal benefits in contexts that require long-term trust and cooperation, such as negotiations and business partnerships. Additionally, the ethical implications of these findings are significant (Coeckelbergh, 2020). By leveraging the adversarial framework, we showed that AI models must be both robust enough to withstand adversarial manipulation and flexible enough to recognize fairness cues and respond accordingly. The framework's ability to simulate adversarial and cooperative scenarios enables testing of how AI systems will perform in real-world contexts where trust, fairness, and adaptability are critical for both success and ethical alignment.

In summary, the adversarial framework offers a novel and effective way to assess the strengths and weaknesses of LLM decision-making. It allows researchers to probe how AI systems balance risk and reward, exploration and exploitation, and stability and adaptability in complex, real-world situations. The findings from both the bandit task and the MRTT illustrate the importance of developing AI systems that can dynamically adjust to new information while safeguarding against adversarial manipulation, which is essential for ensuring the success and ethical alignment of AI systems in diverse applications.

## 5  LIMITATION

While our study provides valuable insights into the decision-making processes and adversarial vulnerabilities of three well-known LLMs, several limitations must be acknowledged. The controlled, structured environments used in our simulations may not fully represent the complexity of real-world scenarios. We focused on two specific tasks, the two-armed bandit task and the MTT, which do not cover all potential decision-making contexts for LLMs. Additionally, the adversarial strategies employed, while effective, might not encompass the full range of real-world tactics. These limitations suggest the need for further research with more diverse scenarios, additional tasks, and broader adversarial strategies to enhance the robustness and applicability of our findings, particularly in the context of AI safety.

## 6  CONCLUSION

This study employed an adversarial framework to investigate the decision-making behaviors of three LLMs, revealing model-specific strengths and vulnerabilities. In the bandit task, GPT-4 and Gemini-1.5 exhibited a bias toward exploitation, which made them predictable and susceptible to manipulation. GPT-3.5, while showing a more balanced approach between exploration and exploitation, was still prone to exploitation in adversarial settings due to its risk-seeking behavior. Gemini-1.5, while showing exploitation tendencies in the bandit task, excelled in the MRTT, where it adapted effectively to both MAX and FAIR adversaries, outperforming the other models in both adversarial and cooperative settings. The adversarial framework proposed in this paper proved to be a powerful tool for testing AI resilience and adaptability in real-world applications such as cybersecurity, finance, and autonomous systems. The findings revealed by the adversarial framework underscore the need for AI models to be not only robust against adversarial manipulation but also flexible enough to respond to fairness cues and changing conditions. They should motivate AI engineers to investigate how to develop better and more robust decision making capabilities which have better strategic flexibility Additionally, integrating interdisciplinary approaches from cognitive science and game theory will be crucial to developing AI systems that not only perform effectively but also align with human values, expectations and ethical standards. By fostering AI that can dynamically adjust strategies and recognize manipulative patterns, we can ensure safer and more reliable applications in critical sectors like healthcare and finance.

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

## A APPENDIX

### A.1 TRAINING THE LEARNER MODEL

The architecture of the learner model is a Recurrent Neural Network (RNN) based on gated recurrent unit architecture (Cho et al., 2014). It is implemented using Tensorflow and the gradients were calculated by automatic differentiation (Abadi et al., 2015). The optimal number of cells and iterations for each experiment are presented in Table 1. In the case of the MRTT experiment, the investments are discretized to five actions corresponding to ranges $0 - 4, 5 - 8, 9 - 12, 13 - 16, and, 17 - 20$.

Table 1: Optimal number of cells and training iterations

| Experiment | #cells in RNN | #training iterations | learning rate |
|---|---|---|---|
| Bandit (GPT-3.5) | 5 | 8600 | 0.005 |
| Bandit (GPT-4) | 5 | 1000 | 0.005 |
| Bandit (Gemini-1.5) | 5 | 2000 | 0.005 |
| MRTT (GPT-3.5) | 3 | 3000 | 0.001 |
| MRTT (GPT-4) | 3 | 5000 | 0.001 |
| MRTT (Gemini-1.5) | 3 | 8000 | 0.001 |

### A.2 TRAINING THE ADVERSARY

The Deep Q-learning algorithm was used for training the adversary in both the bandit task and the MRTT. In the bandit task, the reward for the adversary was the learner model selected the target action. In the MRTT, the reward for the MAX adversary in each trial was the earning amount ($3\times$ investment $-$ repayment), while in the case of the FAIR adversary, the reward was zero in each round except for the last round in which the reward was the negative absolute difference between the gains of trustee and investor over the whole task. The adversary neural network has three fully connected layers with 128, 128, and 4 units, employing ReLU activation functions for the first two layers and a liner activation function for the final layer. Replay buffer sizes of 200,000 and 400,000 were considered. The $\epsilon$-greedy method was used for exploration with $\epsilon \in 0.01, 0.1, 0.2$. Learning rates $10^{-3}, 10^{-4}, 10^{-5}$ were considered for training the adversary using the Adam optimizer. The adversary was simulated against the learner model 200 times and the average bias was calculated to evaluate the performance of the adversary. The optimized combination of parameters for each experiment is shown in Table 2.

Table 2: The hyperparameters for training the adversary

| Experiment | buffer size | epsilon | learning rate | #training iterations |
|---|---|---|---|---|
| Bandit (GPT-3.5) | 400,000 | 0.01 | $10^{-3}$ | 100,000 |
| Bandit (GPT-4) | 400,000 | 0.01 | $10^{-3}$ | 100,000 |
| Bandit (Gemini-1.5) | 400,000 | 0.01 | $10^{-3}$ | 100,000 |
| MRTT_MAX (GPT-3.5) | 400,000 | 0.01 | $10^{-3}$ | 40,000 |
| MRTT_MAX (GPT-4) | 400,000 | 0.01 | $10^{-3}$ | 22,000 |
| MRTT_MAX (Gemini-1.5) | 400,000 | 0.01 | $10^{-3}$ | 84,000 |
| MRTT_FAIR (GPT-3.5) | 400,000 | 0.01 | $10^{-3}$ | 42,000 |
| MRTT_FAIR (GPT-4) | 400,000 | 0.01 | $10^{-3}$ | 70,000 |
| MRTT_FAIR (Gemini-1.5) | 400,000 | 0.01 | $10^{-3}$ | 34,000 |

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
