# OpenReview forum: "Adversarial Testing in LLMs: Insights into Decision-Making Vulnerabilities"
_ICLR.cc/2025/Conference — Submitted to ICLR 2025_

### Official Review · Reviewer_khxo · 2024-10-28

**Soundness:** 3
**Presentation:** 3
**Contribution:** 2
**Rating:** 3
**Confidence:** 4

**Summary:**

This paper examines the decision-making capabilities and adversarial vulnerabilities of LLMs, specifically GPT-3.5, GPT-4, and Gemini-1.5, within structured experimental settings designed to simulate complex human interactions. The study employs an adversarial framework in two decision-making tasks: the two-armed bandit task and the Multi-Round Trust Task (MRTT). In the bandit task, the adversary’s goal was to manipulate the LLM’s preference towards a target action under equal reward conditions. In the MRTT, adversaries were trained either to maximize their own earnings (MAX) or to enhance fairness (FAIR).

**Strengths:**

1. The paper focuses on a good aspect when using LLM agents to solve online decision-making problems with adversarial environments.

2. The paper is overall well-written and easy to understand.

3. The behavioral analysis involving human beings is interesting.

**Weaknesses:**

1. The paper could be strengthened by including more recent and relevant studies, such as

Krishnamurthy, Akshay, Keegan Harris, Dylan J. Foster, Cyril Zhang, and Aleksandrs Slivkins. "Can large language models explore in-context?." arXiv preprint arXiv:2403.15371 (2024).

Despite it is an arXiv article to the best of my knowledge, this is among the first few papers seriously discussing applying LLMs to bandit-related problems.

2. Consider incorporating and evaluating more advanced LLMs, such as GPT-4 Turbo and the o1 model. Analyzing these newer models could provide a deeper understanding of the capabilities and limitations of current LLM technologies in adversarial decision-making contexts.

3. The paper considers a 2-armed bandit problem as a case study. It is necessary to extend the number of arms to general settings. The focus on a 2-armed bandit problem, while insightful, limits the generalizability of the findings. Expanding the experimental setup to include MAB problems with more than two choices could offer a more comprehensive view of LLM behavior in complex decision-making scenarios.

In summary, the authors are recommended to integrate comparisons with standard adversarial MAB algorithms and extend the analysis to more general MAB settings, including evaluations on GPT4 Turbo and o1 models.

**Questions:**

1. How does the proposed methods (for example, Gemini 1.5 based on the proposed framework) compare against classic algorithms such as EXP3 for adversarial bandit problems?

2. Can the performance of LLMs be theoretically guaranteed when facing adversarial environments? Can the proposed framework provide some insights?

---

> ### Author Response · Authors · 2024-11-18
>
> 1. Incorporating More Recent and Relevant Studies
>
> Reviewer Feedback: The reviewer suggested adding more recent and relevant studies, such as Krishnamurthy et al.'s work on LLMs and bandit-related problems.
>
> Response: Yes, a relevant paper we neglected to include. We appreciate the recommendation and will incorporate.
>
> 2. Inclusion of Advanced LLMs (e.g., GPT-4 Turbo, o1 Model)
>
> Reviewer Feedback: The reviewer recommended evaluating newer models, such as GPT-4 Turbo and the o1 model, to deepen the analysis of LLMs' capabilities and limitations.
>
> Response: The rapid evolution of LLMs presents challenges in capturing all the latest developments at the time of writing. When we conducted our research, models like GPT-4 Turbo had not yet been released. Our primary focus was on the development and validation of the adversarial framework, which can be applied to a range of LLMs. We believe future research can leverage this framework to analyse the behaviour and decision-making processes of newer models as they become available. The contribution of the work is independent of the number and range of LLMs used. However, it is a good idea to create an online resource where researchers can see the results of the application of our framework to well-known LLMs and agents. We can update this regularly.
>
> 3. Extension to Multi-Armed Bandit (MAB) Settings
>
> Reviewer Feedback: The reviewer noted that focusing on a two-armed bandit problem limits the generalizability of the findings and suggested extending the analysis to more general MAB problems with more choices.
>
> Response: We appreciate the suggestion to extend the study to MAB settings. However, the primary focus of this study is the development and application of our adversarial framework to evaluate LLM decision-making behaviours. Extending the analysis to MAB with multiple arms would shift the scope of the research and raise questions such as how many arms to include. Additionally, we currently only have human data for the two-armed bandit version, which aligns with our aim to compare LLM behaviour directly with human performance. For future research, applying our framework to multi-armed bandit tasks and other complex decision-making scenarios could be investigated. For the time being the MAB in its current form is sufficient to clearly demonstrate the vulnerabilities in decision making for current technologies.  As these technologies become more sophisticated we can enhance the framework with more nuanced decision making assessments.
>
> 4. Comparison with Standard Adversarial MAB Algorithms
>
> Reviewer Feedback: The reviewer recommended comparisons with classic algorithms such as EXP3 for adversarial bandit problems.
>
> Response: Our focus was on understanding LLM-specific decision-making behaviour and adversarial responses, rather than benchmarking against traditional algorithms. However, we recognize that comparing LLMs with standard adversarial MAB algorithms like EXP3 would offer a useful perspective on their performance. It is not the focus of the paper however, which is the introduction of this novel adversarial framework for investigation of decision making vulnerabilities.  In future work, we will consider including such comparisons to highlight how LLMs differ from classic algorithms in handling adversarial settings and decision-making but the contribution here is to first introduce the idea and demonstrate that it produces meaningful insights.
>
> 5. Theoretical Guarantees of LLM Performance
>
> Reviewer Feedback: The reviewer inquired about the theoretical guarantees of LLMs' performance when facing adversarial environments and insights provided by the framework.
>
> Response: The theoretical analysis of LLMs' performance in adversarial environments is an area that merits deeper exploration. While our current framework provides empirical insights into LLM behaviour, it does not formally establish performance guarantees. That is work for others and the future.

---

> > ### Comment · Reviewer_khxo · 2024-11-25
> >
> > Thank you for answering my questions.

---

### Official Review · Reviewer_r98e · 2024-11-03

**Soundness:** 1
**Presentation:** 1
**Contribution:** 2
**Rating:** 3
**Confidence:** 3

**Summary:**

This work presents a framework to evaluate the decision-making processes of language models under adversarial settings. This work evaluates popular LLMs on two settings – a) a two-arm bandit task and b) a multi-round trust task (MRTT). Results show that LLMs are more prone to exploitation than humans on the bandit task. On the MRTT task, Gemini-1.5 outperforms all baseline methods, notably GPT-4. Overall, the results show that there is still scope for improvement in the decision-making strategies of popular LLMs.

**Strengths:**

1. The problem setting is relevant and interesting. As LLMs scale and improve, it is a generally interesting question to understand their decision-making processes, particularly under adversarial settings.
2. The results, although in toy settings, are noteworthy. It is notable that it is possible to train a powerful adversary, and that an RNN-model can replicate the decision-making processes of a LLM to a certain degree.

**Weaknesses:**

1. **Writing**: Writing is very poor and often redundant. The *abstract* and *conclusion* are longer than needed and do not effectively summarize the work. On the other hand, the *method* section does not build up motivation properly and dives straight into notation without a proper setup of the problem. I also think that a separate *related work* section should be added, and some content should be moved over from the introduction. The current flow of the *introduction* is not smooth. There are also experimental details missing (such as details about the human experiments for the bandit task).

2. **Experiments**: The experimental settings are very simplistic and although the insights are interesting, they might not scale/apply to real-world problem setups.

3. **Method**: Some design decisions are unclear and do not seem optimal. For example, the adversary controls the observation of the LLM but it is unclear what this observation is for the bandit task. Similarly, I am not sure why the reward probabilities for both actions are set to be the same in the bandit task.

**Questions:**

1. What are the details about the human experiments conducted for the two-armed bandit task (how many humans, the setup of the human experiment, etc.,)?
2. What is the observation $o_t$ in the bandit task? How does the adversary manipulate this observation?
3.  *“The adversary assigns rewards to both potential actions with the constraint that each action receives an equal number of potential rewards (25 times)”*. What does this mean? How is the constraint of the number of potential rewards enforced? The paper could really benefit from a better description of the mathematical setup.
4. There should be some optimal way to solve the two-arm bandit problem (using algorithms like UCB). What do those trajectories look like and what does the action distribution look like in this case? That is, is there an optimal action distribution?
5. Why are the reward probabilities for both planets in the bandit task set to be the same? If they are slightly different (eg, 0.24 and 0.26), then the optimal limiting policy is to always select the higher probability planet. In such a setting, the target action of the adversary could be to induce the action of the lower reward planet.
6. Figure 4 (sample simulation plots) have very low readability. I think the visualizations could be made better to show the overall trend of the actor and adversary policies.
7. In the abstract, the bandit task is properly defined but the MRTT task is not. The abstract dives straight into the adversaries trained for MRTT without introducing what MRTT is.
8. It would be interesting to provide some analysis on the learned RNN and how well it is able to model the LLM’s decision-making process.

---

> ### Author Response · Authors · 2024-11-18
>
> 1. Writing and Organization
>
> Reviewer Feedback: The reviewer mentioned that writing is poor and redundant, suggesting that the abstract and conclusion be more concise, and recommended adding a separate related work section. They also noted that the method section lacks motivation and dives into notation too quickly.
>
> Response: Given the multidisciplinary nature of the work and the bringing together of ideas from both AI and cognitive science the paper, in order to be self-contained has a lot of material to cover. This has likely led to what is being perceived as “poor writing”.  We can review and streamline the abstract and conclusion to ensure they are concise and effectively summarize the work. We will add a dedicated "Related Work" section to improve the structure and flow of the introduction.
>
> 2. Experimental Details
>
> Reviewer Feedback: The reviewer requested more information on the human experiments conducted for the two-armed bandit task, including the number of participants and the experimental setup.
>
> Response: We chose not to include the full details of the human data collection process due to the page limit constraints of the submission format. However, we have cited the original study where these details can be found for readers who wish to access the comprehensive methodology. These experiments have been carried out as per standard and well accepted standards in the cognitive science community.
>
> 3. Observation in the Bandit Task and Adversary Manipulation
>
> Reviewer Feedback: Clarification was sought on the nature of the observation in the bandit task and how the adversary manipulates it.
>
> Response: These are key elements to understanding this paper and therefore important for reviewers so they can understand what is actually going on in the experiments. So, the observation refers to the feedback given to the LLM after each action, that is whether or not a reward was received in the bandit task, and how much units the trustee return back in the MRTT. The adversary manipulates LLMs’ behaviour by controlling the feedback provided.
>
> 4. Reward Assignment Constraint
>
> Reviewer Feedback: The statement about assigning an equal number of potential rewards to both actions was unclear, and clarification was requested on how this constraint is enforced.
>
> Response: Again, understanding what is happening here is key to understanding the paper and its contribution. The constraint means that over a fixed number of trials, each action (e.g., choosing between two options) has an equal opportunity to receive rewards. This ensures that neither action is inherently more rewarding in the long run. The adversary strategically allocates these rewards to influence the LLM's decision-making without biasing the task itself.
>
> 5. Use of UCB and Optimal Action Distributions
>
> Reviewer Feedback: The reviewer suggested comparing results to an optimal solution like UCB (Upper Confidence Bound) and analysing the action distributions.
>
> Response: Our study is focused on understanding LLMs' behaviour under adversarial conditions and developing an adversarial framework to evaluate their decision-making processes. The focus is not on optimal action strategies or comparing LLM performance to algorithmic baselines like UCB, but rather on how these models respond to structured adversarial influences. However, as also suggested by the first reviewer, we can consider to include an analysis comparing LLMs decision-making to the exploration strategies adopted in these algorithms to provide a better characterization of exploration mechanisms of the LLMs.
>
> 6. Reward Probabilities in the Bandit Task
>
> Reviewer Feedback: The reviewer questioned why the reward probabilities for both planets were set to the same value and suggested testing with slightly different probabilities.
>
> Response: The equal reward probabilities were chosen to establish a neutral setting where the adversary's influence on the LLM's decision-making could be isolated and observed. It’s also the experimental setting with human participants. We agree that testing with slightly different reward probabilities (e.g., 0.24 vs. 0.26) could add depth to the analysis, showing how the adversary performs when there is an inherent bias. This could be an interesting extension for future work.
>
> 7. Figure Readability
>
> Reviewer Feedback: Figure 4 has low readability, with suggestions to improve the visualizations.
>
> Response: We will revise the visualizations to present clearer trends, possibly through larger plots and more distinct markers for different trajectories.
>
> 8. MRTT Task Introduction in Abstract
>
> Reviewer Feedback: The MRTT task was not properly introduced in the abstract.
>
> Response: We can revise the abstract to include a brief introduction to the MRTT task.
>
> 9. Analysis of the Learned RNN
>
> Reviewer Feedback: The reviewer suggested providing analysis on how well the RNN models the LLM’s decision-making process.
>
> Response: We can provide the loss curve of the RNN model.

---

> > ### Comment · Reviewer_r98e · 2024-11-21
> >
> > Thank you for your response. I personally do not think the writing is because of the multidisciplinary nature of the work -- I think some figures could have been moved to appendix or made more concise to enable more writing space. I also feel that the writing was redundant and important details were missing.
> >
> > I know that the observation and the reward function of the adversary are key details of the paper but I found them difficult to understand when I was reading the paper. I think mathematically defining them in Section 2.2 (for the bandit problem) and Section 2.3 (for MRTT) will increase the clarity of the paper.
> >
> > I would also like to point out that when running human experiments, it is standard to provide detailed experiment conditions (it is okay to place these details in the Appendix if the main paper does not have space).
> >
> > Overall, while this paper studies an interesting and relevant problem, it does not meet the threshold for acceptance in its current state. I have left my score unchanged.

---

> > > ### Author Response · Authors · 2024-11-28
> > >
> > > While we acknowledge that writing clarity is critical, we respectfully disagree that the writing is problematic. Our team includes fluent English speakers who reviewed the manuscript to ensure it was well-articulated and concise. However, we understand that individual perceptions of clarity can vary, especially given the multidisciplinary nature of the work.
> > >
> > > Can you point out which part is redundant and what details are missing? We don't think human experiments and data need to be elaborated in our paper since they have been published in another paper and we have properly cited in our manuscript. Plus we didn't run the human experiment and that's not the focus of this paper. We are mainly looking at LLMs' behaviour.
> > > The two-arm bandit task and the MRTT were described conceptually rather than mathematically because they are standard paradigms in cognitive and behavioral science (that's also why we think the writing issue is because of the multidisciplinary nature of this paper). Providing mathematical definitions could detract from the focus of the paper, which is the behavior of LLMs under adversarial conditions.

---

> > > > ### Comment · Reviewer_r98e · 2024-11-30
> > > >
> > > > - The discussion and conclusion have a lot of redundancies. Lines (495-500) are basically scaffolding that is repeated in the conclusion.
> > > >
> > > > - *"The adversary assigns rewards to both potential actions with the constraint that each action receives an equal number of potential rewards (25 times)"* (Line 179) is unclear to me. Maybe this is because of the multidisciplinary nature of the work, but it is standard in RL to define this more concretely.
> > > >
> > > > - Lines 450-458 (First paragraph of discussion). This paragraph contributes very little technical discussion and is very generic.
> > > >
> > > > - Lines 174-175. The title of this subsection is the bandit task, yet the first sentence mentions MRTT. This kind of structuring breaks flow.
> > > >
> > > > - Line 180: *"This experiment aims to subtly influence GPT’s preferences and evaluate the adversary’s effectiveness under these constraints."* . The bandit experiment is also run on Gemini, this sentence should not say "GPT" but instead say "LLMs".
> > > >
> > > > - The paper has no related work section.
> > > >
> > > > - Both the abstract and conclusion are overly lengthy. This is a subjective opinion: the authors are free to disagree but I believe both can be made much shorter. Personally, I would not mention specific LLMs in the abstract as the primary contribution of the paper is the evaluation framework. (Lines 86-87 mention this)
> > > >
> > > > - Figure 1 caption is way too long. I think there are better ways to present this (maybe breaking it into two figures).
> > > >
> > > > - Figure 4 does not have good interpretability. It is difficult to draw insights from it. There might be better ways to communicate the insights from this figure. (Same for Figure 2B). Again, this might be because of the multidisciplinary nature but I personally found it hard to read.
> > > >
> > > >
> > > > I will maintain my current score

---

### Official Review · Reviewer_zt19 · 2024-11-04

**Soundness:** 2
**Presentation:** 2
**Contribution:** 2
**Rating:** 5
**Confidence:** 4

**Summary:**

This paper presents an  exploration of the decision-making vulnerabilities of Large Language Models (LLMs) by applying an adversarial framework to two classic decision-making tasks: the two-armed bandit task and the Multi-Round Trust Task (MRTT). The authors test three prominent LLMs: GPT-3.5, GPT-4, and Gemini-1.5, comparing their performance to human participants in both tasks.

**Strengths:**

The proposed framework looks interesting. The problem of LLM decision making with adversarial testing is new.

**Weaknesses:**

Prompt Robustness:  LLMs can be very sensitive to the prompt design. Is the observation a product of particular prompt design? The paper has only used a single prompt for both the tasks. A generic framework could be designed where prompts can be varied and new scenarios added to check the robustness of the results. Please create variations of the scenarios to make sure the observations are indeed generic. Please also give the temperature of the LLM used and if possible make more runs to report trends.

In MRTT, it needs to be ensured to conduct the experiment with different range of values to check if they hold before inferring conclusions on LLM behavior .

Lack of understanding of GPT 3.5 exploratory mechanisms : Though evidence accumulation is well studied in humans , it is still relatively new in LLMs. The inference that the switching action of GPT 3.5 underlies robust exploration strategies needs more evidence, though the current adversarial design might not accommodate it. It could just be a greedy mechanism that is immune to this particular choice of adversarial attack instead of any robust exploratory mechanism.

Controlled Environments: While the use of controlled environments allows for rigorous experimentation, it may not fully capture the complexity of real-world decision-making scenarios. Inclusion of some realistic RL scenarios as a case study or an experiment will benefit the paper from an alignment perspective. Also, the capabilities of the adversary can be specified more clearly with applications.

Please improve captions of figures and include at least one open source LLM if possible.

**Questions:**

In  L320 'burned target' is not clear to me. Can u please elaborate.

In L327, it is give "Unlike GPT-3.5, GPT-4 and Gemini-1.5 consistently preferred the target action once established." I dont think this is new information , prior papers have explored LLM decision making and fixations on rewards in a greedy mechanism, but they haven't been cited. Please check the following papers:
Can large language models explore in-context?
Controlling Large Language Model Agents with Entropic Activation Steering

In the original Dezfouli paper from which the experiment design is inspired also has a Go/No Go paradigm, any reason why this wasn't tested?

---

> ### Author Response · Authors · 2024-11-18
>
> 1. Prompt Robustness
>
> Reviewer Feedback: The reviewer noted the potential sensitivity of LLMs to prompt design and recommended testing with varied prompts and different temperature settings for robustness.
>
> Response: We did conduct experiments using variations in prompt design. Our results here demonstrated that minor variations in prompts or even variations of scenarios did not impact the overall performance of LLMs on the decision-making tasks and the efficiency of the corresponding adversaries. In all cases the LLMs were sensitive to adversarial manipulation in decision making.
> We tested using the default temperature values in this study as these represent a real world use scenario. Different temperatures did impact LLMs’ precise behaviour in these decision-making tasks based on our experiment but it did not yield any immunity to manipulation. In any case since the focus of this paper is the introduction of a novel adversarial framework, we didn’t report the results under different temperature settings in this paper.
> 2. MRTT Experiment Range
>
> Reviewer Feedback: The reviewer suggested conducting the MRTT with different ranges of values to confirm that conclusions on LLM behaviour hold under varied conditions.
>
> Response: We appreciate the reviewer's recommendation. We used the standard MRTT setup as employed in cognitive and behavioural science research to maintain consistency with existing human studies. This alignment also allows us to directly compare LLM performance with human benchmarks under established conditions. The primary contribution of the paper is that this type of adversarial framework which has well-known efficacy in human cognitive science studies is translatable into the AI domain and in particular yields insightful results with model LLM-based agent systems.
>
> 3. Exploration Mechanisms of GPT-3.5
>
> Reviewer Feedback: The inference about GPT-3.5's exploratory mechanisms may require additional evidence, as it could be due to simple greedy behaviour rather than robust exploration.
>
> Response: We acknowledge the concern about the characterization of GPT-3.5's exploration mechanisms. This was a speculative remark on our part and not a contribution of the paper. That said, we can  include an analysis comparing LLMs decision-making to more standard exploration strategies (e.g., epsilon-greedy, UCB). This will allow us to distinguish between true exploration and simple greedy behaviours, strengthening our interpretation.
>
> 4. Controlled Environments and Real-World Scenarios
>
> Reviewer Feedback: The paper's controlled environments, while rigorous, might not capture real-world complexity. Including realistic RL scenarios as case studies was recommended.
>
> Response: We appreciate the reviewer’s insight on this matter. It’s important to note that the use of controlled environments reflects the standard methodology in cognitive and behavioural science for investigating human behaviour. Measuring human behaviour in real-world scenarios is challenging, as there is currently no established methodology for conducting such investigations with scientific rigor. Similarly, assessing LLM behaviour scientifically requires controlled, simulated scenarios to ensure valid and reliable measurement. While real-world simulations can offer additional context, there are too many variables that we can’t control, so the controlled tasks used in our study align with best practices for systematically probing decision-making processes. Future work, can address such concerns but we suggest our paper as it stands lays down the important foundational research for such future investigations
>
> 5. Figure Captions
>
> Reviewer Feedback: Improve figure captions
>
> Response: We agree with the recommendation to enhance figure captions for better clarity and will make these revisions accordingly.
>
> 6. Clarification on 'Burned Target' (L320)
>
> Reviewer Feedback: The term 'burned target' was unclear.
>
> Response: The term "burned target" refers to a strategy where the adversary intentionally refrains from rewarding the target action to prevent the model from easily predicting reward patterns. We will ensure this explanation is clearer in the revised manuscript.
>
> 7. Prior Work and Citations
>
> Reviewer Feedback: The reviewer pointed out the need to cite related works on LLMs' exploration and fixation on rewards.
>
> Response: We appreciate the references provided and will include relevant citations. This improves context setting although none of these papers subtract from the contribution of what we are proposing in terms of a new adversarial framework for AI agents decision making.
>
> 8. Go/No Go Paradigm
>
> Reviewer Feedback: The reviewer asked why the Go/No Go paradigm from the original Dezfouli paper was not tested.
>
> Response: The Go/No Go paradigm was not included in our study because it inherently involves motor responses or physical actions—specifically, clicking or refraining from clicking—which LLMs cannot perform.

---

> > ### Comment · Reviewer_zt19 · 2024-12-03
> >
> > Thanks for the response, and sorry for the delay in the response.
> > I still think `showing’ the results for robustness to prompt in the paper/appendix is important. Especially since the paper studies specific prompts like space travel, it is important to show that the observations scale.
> >
> > The finding is indeed interesting and worth exploring. In case the paper does not make it this time, please scale the experiments with diverse and more generic prompts in future. Also recommend to include more LLMs, at least an open source LLM.

---

### Meta-Review · Area_Chair_jJ4D · 2024-12-10

**Metareview:**

This paper studies the vulnerability of LLMs in decision-making under adversarial scenarios, in benchmark settings of the multi-armed bandits and the multi-round trust task. It was then shown that most LLMs are more prone to exploitation than human decision-makers. The overall problem is interesting, relevant, and significant. However, it reached a consensus that the paper still has much to improve in terms of the extensiveness and sufficiency of the experiments, as well as the clarity and rigor of the exposition. I suggest the authors incorporate the feedback and improve the paper for other upcoming ML venues.

**Additional Comments On Reviewer Discussion:**

Several reviewers share the comments that the experiments are not sufficient and large-scale enough to make the case. The authors acknowledge the feedback, but deferred most efforts to future work, and did not make the effort to address the comments by providing more controlled experiments. There were also some concerns regarding the writing/presentation, in terms of clarity and rigor (which I very much agree). The authors acknowledged the feedback but also had some pushback. Overall I think the paper has not cleared the bar for acceptance.

---

### Decision · Program_Chairs · 2025-01-22

Reject